# StressMatic: A Novel Automated System to Induce Depressive- and Anxiety-like Phenotype in Rats

**DOI:** 10.3390/cells12030381

**Published:** 2023-01-20

**Authors:** Joana Martins-Macedo, António Mateus-Pinheiro, Cátia Alves, Fernando Veloso, Eduardo D. Gomes, Inês Ribeiro, Joana S. Correia, Tiago Silveira-Rosa, Nuno D. Alves, Ana J. Rodrigues, João M. Bessa, Nuno Sousa, João F. Oliveira, Patrícia Patrício, Luísa Pinto

**Affiliations:** 1Life and Health Sciences Research Institute (ICVS), School of Medicine, University of Minho, 4710-057 Braga, Portugal; 2ICVS/3B’s—PT Government Associate Laboratory, 4806-909 Braga/Guimarães, Portugal; 3Bn’ML—Behavioral & Molecular Lab, University of Minho, 4710-057 Braga, Portugal; 4Department of Marketing and International Business, University of Vienna, Oskar Morgenstern-Platz 1, 1090 Vienna, Austria; 52Ai—School of Technology, IPCA, 4750-810 Barcelos, Portugal; 6LASI—Associate Laboratory of Intelligent Systems, 4800-058 Guimarães, Portugal; 7Department of Mechanical Engineering, School of Engineering, University of Minho, 4800-058 Guimarães, Portugal

**Keywords:** automated rack, stress-exposure, protocols standardization, preclinical research

## Abstract

Major depressive disorder (MDD) is a multidimensional psychiatric disorder that is estimated to affect around 350 million people worldwide. Generating valid and effective animal models of depression is critical and has been challenging for neuroscience researchers. For preclinical studies, models based on stress exposure, such as unpredictable chronic mild stress (uCMS), are amongst the most reliable and used, despite presenting concerns related to the standardization of protocols and time consumption for operators. To overcome these issues, we developed an automated system to expose rodents to a standard uCMS protocol. Here, we compared manual (uCMS) and automated (auCMS) stress-exposure protocols. The data shows that the impact of the uCMS exposure by both methods was similar in terms of behavioral (cognition, mood, and anxiety) and physiological (cell proliferation and endocrine variations) measurements. Given the advantages of time and standardization, this automated method represents a step forward in this field of preclinical research.

## 1. Introduction

Major depressive disorder (MDD) is currently the leading cause of disability worldwide, according to the World Health Organization [1]. Its symptoms include persistent low mood, anhedonia, guilty feelings, and cognitive dysfunction [2]. Continuous and unpredictable exposure to stress is one of the most prominent precipitating factors for depression [3]. Preclinical models of depression are extensively used to understand neurobehavioral changes in the context of the disease and evaluate the efficacy of novel antidepressant therapies. Depression models are established based on three main validity criteria: face validity (representative phenotype of human symptomatology), construct validity (similar causative features of the pathology), and predictive validity (reversal of symptoms induced by the disease through pharmacological or non-pharmacological approaches) [4,5]. The vast majority of strategies that are used to produce these models are based on stress exposure paradigms, the administration of glucocorticoids, genetic manipulations, and the interaction between gene and environmental factors [4,6,7].

Stress exposure models, in particular, are one of the most widely used as they explore neural mechanisms that underlie depressive-, anhedonic- and anxiety-like paradigms [8,9].

The hypothalamic–pituitary–adrenal (HPA) axis at least partly mediates the above-mentioned behavioral changes. In response to stress, the HPA axis is activated and instructs the endocrine system to release glucocorticoids (corticosterone in rodents, cortisol in humans) to induce a negative feedback mechanism through glucocorticoid receptors and reestablish homeostasis in acute conditions. However, upon chronic stress, the persistent stimulation of the HPA axis can result in an uncontrolled release of glucocorticoids, which leads to neural damage [10,11,12,13]. The unpredictable chronic mild stress (uCMS) model, originally developed by Paul Willner, is one of the most well-characterized and validated models of depression [14,15]. In this model, rodents, typically rats, are exposed during a prolonged period, usually from 4 to 8 weeks to a sequential number of mild stressors in an unpredictable manner. The stressors include alterations in light/dark periods, housing settings, and feeding/drinking habits. The stressor’s intensity level or exposure duration may increase throughout the course of the protocol. uCMS constitutes a valuable model for performing antidepressant drug screening and, consequently, studying drug mechanisms of action, treatment resistance, and disease pathophysiology [14,16]. This model also induces cellular and molecular alterations that are relevant to disclosing the neurobiological context of MDD [17,18].

Despite its translational potential and validity, the uCMS model has been criticized for its lack of reproducibility. Several studies report difficulties in reproducing the uCMS protocol [14]. There are obstacles in recapitulating the details and conditions under which the original data were produced. Additionally, some authors report divergences in protocols, the disparity in the conditions of the animal facilities, and variations in the susceptibility of each individual to the protocol [15]. Moreover, this is a very demanding protocol to implement that requires significant time and intensive labor.

To mitigate these constraints and reproducibility issues, we developed and validated an automated system for uCMS exposure in rodents based on the original uCMS protocol. Herein, we present behavioral and neurobiological evidence to validate this model as we compare it to a manual version of the uCMS.

## 2. Materials and Methods

### 2.1. Animals

Two-month-old male Wistar Han rats weighing 200–250 g (Charles-River Laboratories) were maintained under standard laboratory conditions (12 h light/12 h dark cycles, 22 °C, 55% of relative humidity, and ad libitum access to food and water). Rats (n = 8 per group) were randomly assigned to the following experimental groups: non-stress control (CT), manual stress (uCMS), and automated stress (auCMS). The uCMS group followed the protocol for 6 weeks as previously described and validated [19]. Moreover, a subset of animals was subjected to the novel auCMS protocol. CT animals were handled by the experimenter twice a week for habituation before the behavioral analysis. All procedures were executed in accordance with the EU Directive 2010/63/EU on animal care and experimentation.

### 2.2. Rack Development & Stress Categories

The automated rack was entirely designed and developed by us with the help of mechanical, electronics, and software engineers to enable the performance of the same stressors of a traditional uCMS protocol but in an automated manner. The design is based on a conventional housing rack, employing standard and commercially available cages and water bottles. Each stressor was developed and tested individually to deliver the same type of stimulus and approximate the intensity of the manual uCMS protocol but in an automated manner. Given that not all stressors are prone to automation, they have been separated into three categories: fully automated, partially automated, and manual (Table 1). The protocol was carried out over a 6-weeks period. The equipment is controlled and programmed through a computer allocated to the automated rack; it is from a computer that the scheduled protocols are uploaded, and the intended stressors are selected (Figure 1). Details regarding the different stressors are given in the Appendix A.

### 2.3. Weight Gain Monitoring

Weight was measured every week throughout the experiment to monitor alterations induced by both uCMS and auCMS protocols.

### 2.4. Sucrose Consumption Test (SCT)

Anhedonic behavior was evaluated at weeks 4 and 6 of both uCMS and auCMS protocols through the sucrose consumption test (SCT). Sucrose preference was also assessed at the baseline (1-week habituation period prior to uCMS) to set homogenous experimental groups. The SCT consisted of presenting two previously weighed drinking bottles, one filled with water and the other filled with 2% (m/v) sucrose solution, for 1 h. Rats were food and water-deprived for 12 h before the test. The individual preference for sucrose was calculated using the following formula: sucrose preference = (sucrose intake)/(sucrose intake + water intake) × 100, as previously described [20]. SCT was performed in the nocturnal activity period (starting at 8:30 p.m.), conducted 24 h after the third trial of the sweet drive test (SDT); following this trial and until 12 h preceding the SCT, animals were allowed to feed freely.

### 2.5. Sweet Drive Test (SDT)

SDT was employed to further assess anhedonia. Rats were exposed and habituated to sweet pellets (3.77 kcal/g; Honey Cheerios^®^; Nestlé Portugal S.A., Linda-a-Velha, Portugal) one day before the first trial. Animals were food deprived for 12 h before the trial test, and during the light period, suspending stress exposure was employed in the uCMS and auCMS groups. The SDT apparatus is comprised of a black acrylic enclosed arena (dimensions: 82 cm × 44 cm × 30 cm), divided by transparent acrylic perforated walls into three closed chambers and one pre-chamber where the animal was initially placed. Each animal crosses a trap door connected to a middle chamber, allowing it to explore both the right and left chambers of the apparatus for 10 min. A total of 20 regular food pellets (3.60 kcal/g; Certificate standard diet 4RF21; Mucedola, S.R.L., Settimo Milanese, Italy) were placed in a corner of the left chamber, while 20 sweet pellets were positioned in a corner of the right chamber. Additionally, both chambers were equipped with ultrasound microphones to record the animal’s ultrasonic vocalizations (USVs) during trials. Specifically, we placed ultrasound microphones (CM16/CMPA, Avisoft Bioacoustics, Glienicke/Nordbahn, Germany) sensitive to 10–200 KHz frequencies, 20 cm above the ground, which were connected via an Avisoft UltrasoundGate 416H (Avisoft Bioacoustics) to a PC, to record the animals’ vocalizations. Importantly, all 50 KHz USVs (typically associated with positive pleasurable experiences) were identified by the software and analyzed by the experimenters. At the end of the trial, sweet pellet preference levels were calculated as follows: preference for sweet pellets (%) = Consumption of Sweet Pellets (g)/Total Food Consumption (g) × 100, as previously described [21].

### 2.6. Elevated Plus Maze Test (EPM)

The assessment of anxiety-like behavior was conducted using the EPM test, performing a single 5 min trial, as previously described [19]. Anxiety-like behavior was inferred by the percentage of time spent with the open-arms.

### 2.7. Novelty Suppressed Feeding Test (NSF)

Behavioral traits of anxiety were also evaluated by the NSF test. Following a food deprivation period of 18 h, the rats were placed in an open-field box, with a food pellet positioned in the center, as previously described [19]. After reaching the pellet, animals were transferred to their home cage and allowed to feed for 10 min. The latency to feed in the open-field arena was taken as a proxy of anxiety-like behavior, and the food consumption in the home cage was used as a measure of appetite drive.

### 2.8. Novel Object Recognition Test (NOR)

The NOR test was performed to evaluate cognitive function. Firstly, rats were habituated to the testing box for 8 min. On the following day, they were allowed to explore two indistinguishable objects positioned in the test field for 10 min (sample phase). Twenty-four hours later (long-term memory), the animals returned to the arena for 3 min, where one of the objects had been replaced by a new one (choice phase). On the last day, animals were evaluated for short-term memory by measuring the time it took to explore a new object one hour after a new sample phase. Importantly, the familiar and new objects were different in color, shape, size, and texture. The testing box was cleaned with 10% ethanol between the trials to avoid odor cues. All sessions were video recorded, and the time spent exploring each object was assessed manually by experimenters unaware of the experimental groups. Recognition memory was expressed by the discrimination index (D), which was defined as D = (time of exploration of the novel object—time of exploration of the familiar object)/total time of exploration.

### 2.9. Forced Swim Test (FST)

Depressive-like behavior was assessed in the FST. After 24 h following a 5 min habituation session, test trials were performed. Animals were placed in water-filled transparent cylinders (25 °C; 50 cm depth) for 5 min. All trials were videotaped, and the immobility time was measured. An increase in immobility time was defined as a proxy of depressive-like behavior.

### 2.10. Corticosterone Levels Measurements

The levels of corticosterone were assessed in the blood serum, which was collected through tail venipuncture, and quantified via a commercially available ELISA kit (Abcam). Samples were obtained at the end of the stress protocols between 8 a.m. and 9 a.m. (nadir, basal levels) and between 8 p.m. and 9 p.m. (zenith; peak).

### 2.11. Immunostaining Procedures

The rats were deeply anesthetized and transcardially perfused with 0.9% NaCl and 4% of paraformaldehyde (PFA). Brains were then removed from the skull, postfixed in 4% PFA, and cryoprotected in a 30% sucrose solution. Coronal sections (40 µm thickness per section) extending over the entire length of the hippocampal formation were obtained using a vibratome (Leica VT 1000 S, Leica Microsystems Nussloch GmbH, Nussloch, Germany). Those containing the dorsal hippocampal dentate gyrus (DG) were stained to assess cell proliferation (BrdU; rat, 1:100; Abcam; ref. ab6326). The density of BrdU+ cells in the DG was normalized by the corresponding area of the DG. BrdU counts were performed in 18 sections from 6 rats (3 sections/rat) for each experimental group. The observer was unaware of the experimental groups. A confocal microscope (Olympus FluoViewTM FV1000, Hamburg, Germany) and an optical microscope (Olympus BX51, Germany) were used for the analysis.

### 2.12. Statistical Analysis

Statistical analyses were performed using GraphPad Prism software. After performing the normality tests (Appendix A), data were subjected to the appropriate statistical tests. One-way analysis of variance (ANOVA) was used to assess differences between the CT group and the stress-exposed groups. A *t*-test was performed to assess differences between both stressed groups, and a repeated measure two-way ANOVA was conducted for the statistical analysis of weight gain and plasma corticosterone levels. Bonferroni’s post hoc multiple comparisons were used to determine differences among the groups. All statistical analyses are reported for each test. In cases where normality was not verified, we applied the Kruskal–Wallis Test. Statistical significance was set at *p* < 0.05. Outliers were calculated using the ROUT method (Q = 1%) from GraphPad Prism software.

## 3. Results

### 3.1. Behavioral Validation of the auCMS Protocol

To validate the automated rack (auCMS) as a stress inducer in rats, we tested its efficacy for inducing core signals of a depressive-like behavior through different behavioral paradigms. The uCMS exposure model is recognized to induce the core signals of depressive-like behaviors in rodents [19]. As a first approach, in the fourth week of stress exposure, we aimed to evaluate if the SCT was able to discriminate the impact of uCMS and auCMS in anhedonic behavior during the chronic period of exposure (Figure 2A,B). In the test paradigm, auCMS- and uCMS-exposed animals presented lower sucrose preference levels than the control animals in the SCT. Although uCMS-exposed animals evidenced a statistically significant decreased preference for sucrose solution over water solution (≈77%, *p* = 0.0287) (Figure 2B), the difference between the control animals and auCMS-exposed was not statistically significant (≈81%, *p* = 0.1309). We also monitored the animals’ weight throughout the protocol. We observed that although all groups increased their weight with time, stress exposure, for both uCMS and auCMS protocols, induced a significant reduction in total body weight gain when compared to non-stressed animals (controls) (Figure 2C, F(2, 147) = 117.7; *p* < 0.0001). 

All further behavioral dimensions were evaluated during the sixth week of stress exposure (Figure 3A). Herein, we repeated the SCT evaluation and observed that both the uCMS and auCMS groups showed a significantly decreased preference for the sucrose solution over the water solution (_uCMS_ ≈ 92%, *p* = 0.0046; _auCMS_ ≈ 94%, *p* = 0.0416), in comparison to the control group (preference values ≈ 98%) (Figure 3B).

To obtain a multi-parametric analysis of anhedonic behavior, we also assessed sweet pellet preference through the SDT test [21]. In this paradigm, preference for the sweet pellets was complemented with the simultaneous recording of 50 KHz ultrasonic vocalizations (USVs), which are recognized as “positive” vocalizations in rodents (Figure 3C). We observed that uCMS- and auCMS-exposed animals (p_uCMS_ = 0.0009, p_auCMS_ = 0.0102) evidenced a significantly decreased preference for sweet food pellets over regular pellets (_uCMS_ ≈ 32%; _auCMS_ ≈54%) in comparison to the control animals (preference values ≈ 100%) (Figure 3C1). Regarding the USVs, stress-exposed animals also presented a reduction in the number of 50 KHz “positive” vocalizations during the test (Figure 3C2) when compared to the controls (p_uCMS_ < 0.0001, p_auCMS_ = 0.0162). Moreover, and despite no significant differences between the number of incursions on both left and right food chambers, stress-exposed animals revealed an overall reduced exploratory behavior in comparison to the control animals (Figure 3C3, F(2, 42) = 5.788, *p* = 0.0060).

To discriminate anxiety-like phenotypes between the controls and stress-exposed groups, we performed the EPM and the NSF tests. In the EPM, we observed that uCMS- and auCMS-exposed animals (_uCMS_ ≈ 11%, *p* = 0.0415; _auCMS_ ≈ 10%, *p* = 0.0314) spent significantly less time in the open-arms than the control animals (≈25%), which is indicative of anxiety-like behavior (Figure 3D). Concomitantly, stress-exposed animals (_uCMS_ ≈ 239 s, *p* = 0.0014; _auCMS_ ≈ 240 s, *p* = 0.0014) also exhibited a significantly decreased latency time to reach the pellet in the NSF when compared to the control animals (≈58 s) (Figure 3E1). It is noteworthy that the animals in the different groups ingested similar amounts of food (Figure 3E2, H(3) = 1.205, *p* = 0.5474). Regarding depressive-like behavior, stressed-exposed animals (_uCMS_ ≈ 68 s, *p* < 0.0001; _auCMS_ ≈ 39 s, *p* = 0.0016) exhibited significantly higher immobility time in the FST (≈15 s) than the control group (Figure 3F).

Stress exposure also dysregulates cognitive functions that depend on the structural integrity of the hippocampus, prefrontal cortex, and reciprocal connections between these two regions. Thus, we performed the NOR test to assess short- and long-memory. We observed that stress-exposed groups (_uCMS_ ≈ −0.52, *p* = 0.0087; _auCMS_ ≈ −0.52, *p* = 0.0161) exhibited a reduced discrimination index in a short-term memory task compared to the control animals (≈−0.08) (Figure 3G1). Considering long-term memory, stress-exposed groups (discrimination index_uCMS_ ≈ −0.02, *p* < 0.0001; discrimination index_auCMS_ ≈ 0.02, *p* < 0.0001) also presented a reduced discrimination index compared to the control group (discrimination index ≈ 0.45) (Figure 3G2). These results suggest that cognition was significantly and identically affected by uCMS and auCMS exposure, which was reflected by a decreased discrimination index.

### 3.2. Endocrine Stress-Induced Changes

Additionally, we sought to assess whether stress-induced behavioral alterations were accompanied by a disruption in the normal corticosterone serum levels (Figure 4A1,A2). Thus, we analyzed plasma corticosterone (CORT) levels in the blood serum of all experimental groups at two time points: morning (8 a.m., Nadir, basal) and evening (8 p.m., Zenith, peak). In the fourth week of stress exposure, the control (non-stressed) animals displayed higher CORT levels at the zenith compared to nadir, as expected (U = 6, *p* = 0.0093). In contrast, uCMS- and auCMS-exposed animals showed no differences in CORT levels between nadir and zenith (t_uCMS_(16) = 0.8083, p_uCMS_ = 0.4307; t_auCMS_(21) = 1.922, p_auCMS_ = 0.0683), suggesting a disruption in the HPA axis in both experimental groups (Figure 4A). In the sixth week of the protocol, similar observations were detected. While the control animals maintained higher CORT levels at the zenith compared to nadir (t(34) = 4.476, *p* < 0.0001), uCMS-exposed animals displayed higher CORT levels at nadir than at the zenith (t(55) = 2.538, *p* = 0.0140) and auCMS-exposed animals exhibited similar CORT levels at nadir and zenith (t(26) = 0.5255, *p* = 0.6037) (Figure 4A1,A2).

### 3.3. Cellular Proliferation in the Hippocampus

We sought to dissect the impact of manual and automatic chronic stress exposures in the modulation of neural plasticity in the adult hippocampus, exploring its effects on cellular proliferation (BrdU+ cells) in the dentate gyrus. The analysis of cell populations through BrdU staining (Figure 4B) revealed that the number of BrdU^+^ cells was reduced in both uCMS- and auCMS-exposed animals when compared to the controls (F(2,49) = 5.349, *p* = 0.0079; p_uCMS_ = 0.0060; p_auCMS_ = 0.0381). This reduction in BrdU+ cells confirms that auCMS causes a typical decrease in newly born proliferating cells in the hippocampal neurogenic niche observed upon manual protocols of uCMS-exposure.

## 4. Discussion

Animal models of human diseases are designed to replicate the phenotype and pathophysiology of the disease and, by doing so, allow for the testing of novel therapeutic approaches. In the case of MDD research, the generation of valid and effective animal models has been an ongoing and challenging task for many years.

There are several animal models of depression, including those based on stress exposure, (bio)chemical manipulations, genetic alteration, or even derived from lesions that are useful for studying different aspects of the disease [5,7,14,15,22,23,24,25,26,27,28,29,30,31]. Because stress is one of the most potent precipitating factors of depression, animal models of depression based on stress exposure, namely uCMS, are one of the most widely used [14,32]. Yet, poor reproducibility across laboratories due to a lack of protocol standardization, such as divergences between protocols, differences in housing conditions, and influence from the operator, is widely reported [31]. Additionally, from the operational point of view, this model is very time-consuming and demanding [31], requiring very well-trained and experienced human resources. In parallel, an ever-increasing awareness about the need for standardizing scientific protocols and tools has prompted the scientific community to develop automated and robotic approaches, namely for preclinical research [33].

Having an awareness of the operational struggles and attempting to decrease the sources of variability in the uCMS protocol, we developed and built a system capable of performing the uCMS protocol in a standardized and automated manner. Here, we validated this automated (auCMS) protocol and compared it with the behavioral, physiological, and neuroplastic changes promoted by a manual (uCMS) protocol. We observed that the automated stress (auCMS) induced the same behavioral deficits that resulted from a manual protocol (uCMS), namely anxiety- (EPM and NSF) and depressive-like (FST) deficits, as well as both anhedonic (SCT and SDT) and cognitive impairments (NOR). Regarding the SCT, even though in the fourth week of stress exposure, the auCMS-exposed animals could not express anhedonic impairments, the sucrose preference was still reduced compared to the controls. At the sixth week time point, and despite the relatively high levels of sucrose preference in the uCMS and auCMS-exposed animals (~90%), both presented a significantly decreased preference when compared to the CT group and with their baseline levels (~99%). This indicates a mild impact of the stress protocol on hedonic behavior, as measured in this behavioral paradigm. We also performed the SDT as a complementary test since this new approach was developed to assess the same behavioral domain but with higher sensitivity, providing a valuable tool to accurately characterize anhedonic behavior in animals chronically exposed to stress [21].

As a consequence of chronic exposure, animals displayed HPA-axis hyperactivity, leading to the deregulation of glucocorticoid secretion patterns in the blood and promoting an alteration in the circadian regulation of corticosterone secretion [34]. Thus, corticosterone measurement is of the utmost importance to assess the efficacy of stress-exposure protocols [21,22]. Herein we showed that both automated and manual stress-exposure protocols were able to induce significant differences in corticosterone concentration, namely by reverting the effects observed in the controls and by promoting a peak at day and low levels at night, thus confirming the successful chronic stress induction through both auCMS- and uCMS-exposures.

Previous studies have also shown that the uCMS protocol impacts cellular proliferation by reducing the generation of newly born cells in the hippocampal dentate gyrus [35,36]. In our study, when compared to the controls, animals exposed to both uCMS and auCMS protocols displayed decreased levels of BrdU+ cells in the adult dentate gyrus, indicative of impaired cellular proliferation.

Overall, the validation criteria after auCMS exposure showed robust results, with the stress-exposed animals presenting a similar response to the manual protocol—indicating that this new automated rack is a valid and promising tool for preclinical research.

With the implementation of this automated protocol, we decreased labor intensity, assuring less variability in the exposure to stressors and reducing the need for the manipulation of the animals. These certainly contribute to overcoming the reported limitations of the uCMS protocol related to inter-experimenter variability and allowing the standardization of the uCMS protocol between batches and across laboratories.

Robust animal models are crucial to advancing research in the field of health, particularly for the improvement of the current research approaches. Thus, this automated system is a step forward to the global implementation of this widely used model and can serve as a powerful contributor to the better and more reliable screening of novel pharmaceutical compounds.

## Figures and Tables

**Figure 1 cells-12-00381-f001:**
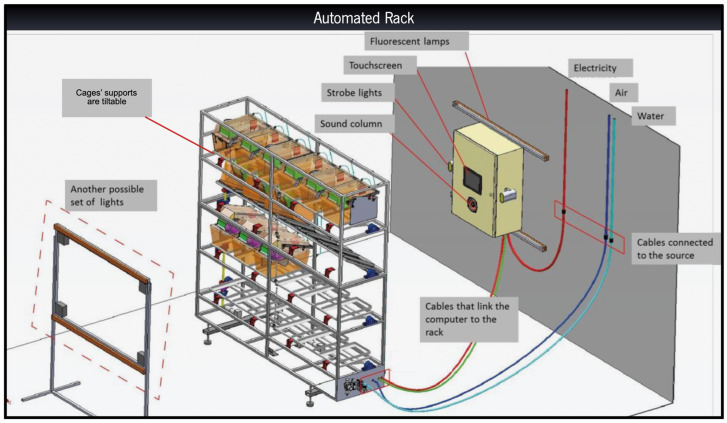
Representative scheme of the automated rack. The design is based on a traditional rack and the whole system is programmed and controlled via a computer allocated to the automated rack, allowing the scheduling of protocols and specific stressors.

**Figure 2 cells-12-00381-f002:**
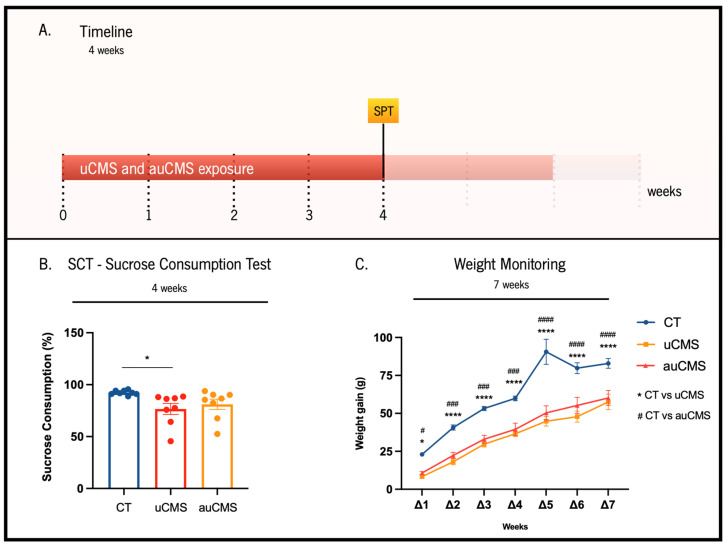
Effects of uCMS- and auCMS exposure on anhedonic behavior at the fourth week of the protocol and weight variation during the experimental protocols. Experimental timeline (**A**). Assessment of anhedonic behavior through the SCT at the fourth week of the protocol (**B**). Weekly monitoring of animals’ weights during the whole protocol of stress exposure to monitor alterations induced by both uCMS and auCMS protocols (**C**). Abbreviations: CT—control; uCMS—manual stressed animals; auCMS—automated stressed animals; Δ—weight variation at a specific week after beginning the stress protocol; In figure (**C**),* means comparison between CT vs. uCMS and # means comparison between CT vs. auCMS over 7 weeks. Data are presented as mean ± SEM. * *p* < 0.05; **** *p* < 0.0001; ^#^ *p* < 0.05; ^###^ *p* < 0.001; ^####^ *p* < 0.0001. n = 8 animals per group.

**Figure 3 cells-12-00381-f003:**
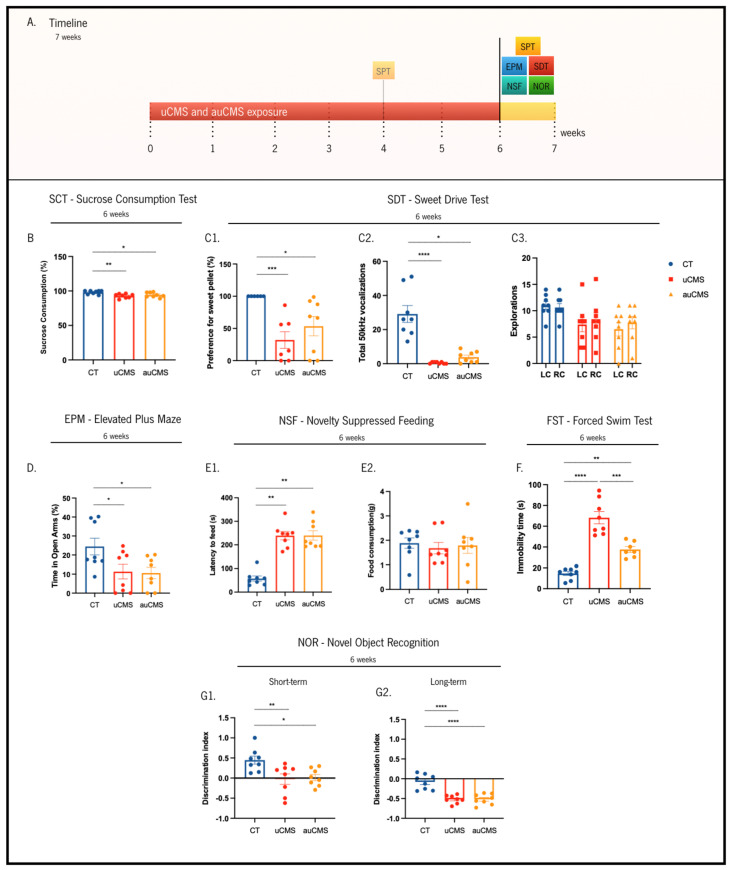
Effects of uCMS- and auCMS exposure in different behavioral dimensions at the sixth week of the protocol. Experimental timeline (**A**). Assessment of anhedonic behavior through the sucrose consumption test (SCT) and (**B**) The sweet-drive test (SDT) (**C1**–**C3**). Assessment of anxiety-like behavior through the elevated plus maze (EPM) (**D**) and the novelty suppressed feeding (NSF) tests (**E1**,**E2**). Assessment of depressive-like behavior through the forced swimming test (FST) (**F**). Assessment of cognitive impairments through the novel object recognition (NOR) test, both at short-term (**G1**) and long-term (**G2**). Abbreviations: CT—control; uCMS—manual stressed animals; auCMS—automated stressed animals; LC—left chamber (sweet pellet); RC—right chamber (regular food). Data are presented as mean ± SEM. * *p* < 0.05; ** *p* < 0.01; *** *p* < 0.001; **** *p* < 0.0001. n = 8 animals per group.

**Figure 4 cells-12-00381-f004:**
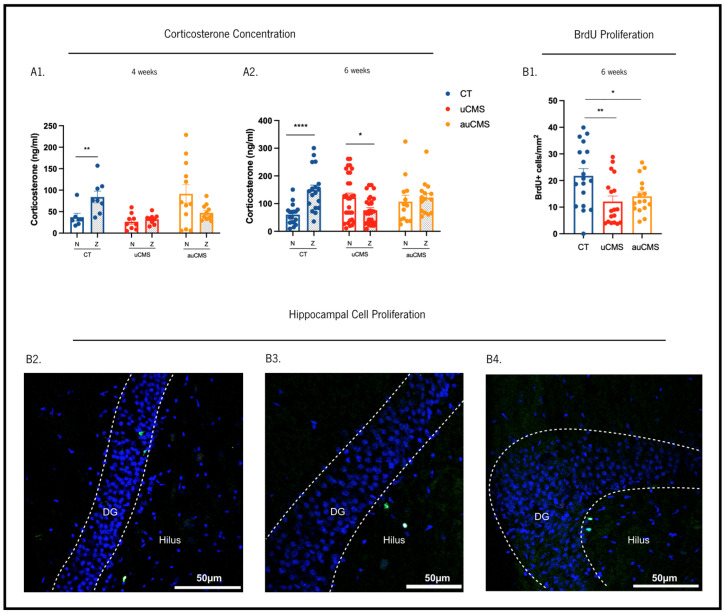
Effects of uCMS- and auCMS exposures in different physiological dimensions. Measurement of corticosterone serum levels on the fourth (**A1**) and sixth week (**A2**). Assessment of hippocampal dentate gyrus cell proliferation through BrdU immunostaining (**B1**). Micrographs depicting examples of BrdU labeled cells (BrdU-positive cells are labeled in green and cell nuclei are labelled with Dapi in blue) in the hippocampal dentate-gyrus of control (**B2**), uCMS-exposed (**B3**) and auCMS-exposed (**B4**) animals. Abbreviations: CT—control; uCMS—manual stressed animals; auCMS—automated stressed animals. Data are presented as mean ± SEM. * *p* < 0.05; ** *p* < 0.01; **** *p* < 0.0001. n = 7–18 samples per group. Scale bar = 50 µm.

**Table 1 cells-12-00381-t001:** Categories of stressors of the automated rack are divided into three groups.

Fully automated	A.	Tilted cage (approximately 45°)
	B.	Housing on damp bedding during the night
	C.	Overnight illumination
	D.	Inverted light/dark cycle
	E.	Exposure to strobe lights
	F.	Startle noise
Manual	G.	Food deprivation followed by exposure to inaccessible food
	H.	Water deprivation followed by exposure to an empty bottle
	I.	Overcrowding
	J.	Cage switch
Partially automated	K.	Confinement to a restricted space

## Data Availability

The data presented in this study are available on request from the corresponding author.

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
