# Peer review of "StressMatic: A Novel Automated System to Induce Depressive- and Anxiety-like Phenotype in Rats"

_cells, 2023, doi:10.3390/cells12030381_

Round 1

Reviewer 1 Report

1. Method to measure ultrasonic vocalizations (USVs) should be mentioned in Materials and Methods part.

2. In section 2.6, the number of entries in closed arms was taken as an indicator of locomotor activity. This result should be included in Results part.

Reviewer 2 Report

The article is focused on the validation of the chronic mild stress model by Willner with the help of an automated system. The model of chronic mild stress has existed for a very long time, but it does require a lot of activity from the experimenter. Automating the protocol would be helpful, although it is not a very actual task. 

1.Did the same rats do all the tests? This is too much for 1 animal. How did you exclude the influence of the tests on each other?  

2.The sucrose preference index  is very high. 90% means that the rats do not have anhedonia at all. A sucrose preference index below 65 is considered an indicator of anhedonia. Maybe these rats have not developed depressive-like behavior? In the Wilner model it is not formed in all animals. Did you exclude animals with high anhedonia. if you did not exclude, then why?

3. what is the reason for comparing corticosterone levels within the same group but not between groups? I recommend adding this comparison as well as a discussion of it in the manuscript. 

Reviewer 3 Report

The authors of “StressMatic: A novel automated system to induce depressive- and anxiety-like phenotype in rats” wrote about a new system to automate a common procedure to model depression and anxiety in rodents. They describe many validation experiments to show that the new automated procedure is like the labor-intensive manual version. They conclude that there is no difference in terms of behavioral or biochemical characteristics of the rats from manual vs. automatic procedures.

I find the paper very well written and easy to understand. Furthermore, the experiments are well chosen to support the aims of the authors. I think there could be a large impact on the field as it would help to standardize and better compare results from different labs. However, this is also the weakness of the paper. I would not be able to replicate the results of this paper or use their automated procedure as it is poorly described. As such, there are no statements on data availability or code used to produce the data in the article. There is also no description of each of the specific manipulations done to the rats or what it means for a task to be automated or semiautomated. Please describe in detail or share the code used to automate the 4 and 6 weeks long exposures to stressors described in the paper.

Other minor issues:

1.     Please report the outcomes of the normality tests or use non-parametric tests as some data looks bimodal (For example, Figure 3D).

2.     Could you expand on the counting of BrdU cells. How many sections from how many rats were used? Please include an example image from each group. Please include at least the catalog number of the materials used.

3.     Please explain the reason to select 4 and 6 weeks timepoints for the different experiments.

4.     Please summarize in a table the number of animals used for each condition in each test. Explain if any animal were excluded from a specific test.

Round 2

Reviewer 2 Report

The authors responded to my comments. Therefore, I kindly ask the authors to add information to the text that the sucrose preference 90 percent does not indicate anhedonia.
